

# Analysis of the performance of Faster R-CNN and YOLOv8 in detecting fishing vessels and fishes in real time

Lotfi Ezzeddini[1,2], Jalel Ktari[3], Tarek Frikha[1,4], Naif Alsharabi[5,6], Abdulaziz Alayba[5], Abdullah J. Alzahrani[5], Amr Jadi[5], Abdulsalam Alkholidi[7] and Habib Hamam[7,8,9,10]

[1] DES Unit, FSS, University of Sfax, Sfax, Tunisia
[2] Higher Management Institute of Gabes, University of Gabes, Gabes, Tunisia
[3] CES Lab, ENIS, Sfax, Tunisia
[4] Computer Sciences and Applied Mathematics Department, ENIS, Sfax, Tunisia
[5] College of Computer Science and Engineering, University of Ha'il, Ha'il, Saudi Arabia
[6] Computer Science Department, College of Engineering and Information Technology, Amran University, Amran, Yemen
[7] Faculty of Engineering, Université de Moncton, Moncton, NB, Canada
[8] International Institute of Technology and Management, Commune d'Akanda, Libreville, Gabon
[9] Bridges for Academic Excellence, Centre-Ville, Tunisia
[10] University of Johannesburg, School of Electrical Engineering, Department of Electrical and Electronic Engineering Science, Johannesburg, South Africa

Corresponding author
Naif Alsharabi, n.sharabi@uoh.edu.sa

## ABSTRACT

This research conducts a comparative analysis of Faster R-CNN and YOLOv8 for real-time detection of fishing vessels and fish in maritime surveillance. The study underscores the significance of this investigation in advancing fisheries monitoring and object detection using deep learning. With a clear focus on comparing the performance of Faster R-CNN and YOLOv8, the research aims to elucidate their effectiveness in real-time detection, emphasizing the relevance of such capabilities in fisheries management. By conducting a thorough literature review, the study establishes the current state-of-the-art in object detection, particularly within the context of fisheries monitoring, while discussing existing methods, challenges, and limitations. The findings of this study not only shed light on the superiority of YOLOv8 in precise detection but also highlight its potential impact on maritime surveillance and the protection of marine resources.

## INTRODUCTION

Maritime surveillance is of crucial importance in preserving fisheries resources and combating overfishing, a pressing threat to global marine ecosystems. Nevertheless, the accurate detection of fishing vessels and schools of fish at sea remains a major challenge, due to the inherent complexity of the marine environment, variable weather conditions and the need for real-time monitoring. To address this challenge, this article explores two of the most promising methods in computer vision: Faster R-CNN (Region-based Convolutional

Neural Network) and YOLOv8 (You Only Look Once version 8). Our main objective is to evaluate their respective performances in detecting fishing vessels and fish using real maritime surveillance data. We defined three key objectives: evaluate the effectiveness of Faster R-CNN and YOLOv8 in maritime object detection, compare their performance in terms of accuracy, execution speed and robustness to changing environmental conditions marine, as well as analyze detection errors and identify areas requiring improvement.

The selection of Faster R-CNN and YOLOv8 was based on their prominence in the object detection literature and their demonstrated effectiveness in various domains. However, we acknowledge that there are other state-of-the-art models available. The decision to focus on these two models was made to provide a comparative analysis while keeping the scope manageable.

Faster R-CNN demonstrates high accuracy in detecting fish, making it suitable for precise identification tasks. However, it suffers from slower processing speeds, increased computational complexity, and limited scalability. Additionally, it may encounter challenges in detecting small or densely packed fish due to its architecture. Moreover, Faster R-CNN requires significant training time, which can be a drawback in dynamic environments.

In contrast, YOLOv8 excels in real-time processing, offering efficient fish detection capabilities with minimal delay. Despite its speed, YOLOv8 may encounter difficulties in detecting small or densely packed fish and exhibit limited localization precision.

In the rest of this article, we will start by presenting a detailed overview of the Faster R-CNN and YOLOv8 methods, followed by a section on the literature review in marine object detection. We will also describe our methodology in detail, including data preparation, model architecture, training, and evaluation metrics. Finally, we will present the results of our experiments in section IV, before entering into an in-depth discussion of the advantages, disadvantages, implications and avenues for future research in Section 'Discussion'. The conclusion, summarizing the main lessons of this study, will close our article in section VI.

## RELATED WORK

### Fish detection

Object detection in computer vision is a fundamental task that has been extensively researched over the past decades. This discipline aims to identify and locate specific objects in images or video sequences (*Wang & Xiao, 2023*). It has varied applications, ranging from face recognition to the detection of autonomous vehicles. Several approaches have been developed to address this problem, including feature-based methods such as SIFT and HOG descriptors, as well as approaches based on convolutional neural networks (CNN).

*Almero et al. (2020)* invented Faster R-CNN, and *Rosales et al. (2021)* also used Faster R-CNN and deep transfer learning to realize K-complex detection in EEG waveform images, which are more accurate, faster, and very close to real-time performance.

Fish detection using imaging technologies and computer vision systems is essential for fish monitoring and meeting growing global demands. A hybrid approach (*Wang & Xiao, 2023*) combines a classification tree and an artificial neural network to

solve this problem. This hybrid model achieves 93.6% training and 78.0% testing accuracy, providing a competitive solution. A Faster R-CNN (*Weihong et al., 2023*) was employed to detect fish with a high accuracy of 99.95% and an average IoU of 0.7816.

Recent research (*Ben Tamou, Benzinou & Nasreddine, 2021*) improved the Faster R-CNN model for marine organism detection. Improvements include the use of Res2Net101, OHEM algorithm, GIOU and Soft-NMS, leading to a significant performance improvement with an mAP@0.5 of 71.7%. Another study *Zhao (2023)*, *Prasetyo, Suciati & Fatichah (2020)* proposed *in situ* detection of underwater jellyfish by improving the Faster R-CNN model. This improved method provides higher training speed and higher accuracy, opening new perspectives for marine ecosystem management. *Adiwinata et al. (2020)* used Faster R-CNN for fish species classification, achieving an accuracy of 80.4%. This research contributes to the conservation of marine species. *Reddy Nandyala & Kumar Sanodiya (2023)* explored the use of YOLOv5 and YOLOv8 with synthetic data for underwater object detection. YOLOv8 has proven to perform well in real-world conditions, which is crucial for underwater environments.

*Patro et al. (2023)* introduced an innovative underwater fish detection system using YOLOv5-CNN, with an average accuracy of 0.86, promising automated monitoring of aquatic resources. In *Büyükkanber, Yanalak & Musaoğlu (2023)* the author presented a model for detecting fish with abnormal behavior, using YOLO v8 and Deep Sort, improving detection despite fish occlusion. *Han et al. (2021)* studied the performance of YOLO and Mask R-CNN for fish head and tail segmentation, showing that YOLO slightly outperformed Mask R-CNN in terms of precision and recall. Sirisha et al. analysis of the YOLO architecture and its variants, highlighting their performance in object detection. YOLO demonstrates state-of-the-art performance in terms of accuracy, speed and memory consumption, but encounters limitations such as detection of small objects and sensitivity to aspect ratios. Despite the importance of this work, it needs to be improved if it is to be applied to surveillance. While this study worked on bases for people, dogs, cows, trains, cars and motorcycles, our approach has been to work on the maritime domain for boats and fish. The use of R-CNN with YOLO was to try and compare the two approaches to try and fill the possible gaps in YOLO V8. *Sirisha et al. (2023)*

This work is the result of collaboration between us the various academic researchers and the Tunisian Ministry of Agriculture and Maritime Fisheries. The database was provided by the partner (*Ktari & Frikha, 2024*) and we used it to obtain the various results.

This work shows significant advances in fish detection using Faster R-CNN and YOLO, contributing to the monitoring of underwater environments and the conservation of marine species. The main objective was to develop a Poof of Concept for an intelligent system for detecting fish and boats. In Tunisia, more and more problems are caused by pollution and the over-exploitation of marine resources. Within the framework of this project, the aim was to use the data proposed by the partner to propose an intelligent approach. We applied Faster R-CNN, YOLOv8 for fish and vessel detection.

## Ship detection

Vessel detection from remote sensing images is crucial in various maritime surveil-lance applications, such as maritime traffic control, combating illegal fishing, and security. Deep learning methods, including YOLOv8, have shown promising results in improving vessel detection from remote sensing images (*Hu et al., 2021*). Additionally, improvements have been made to efficiently detect ships at small scales using attention mechanisms and a new feature pyramid structure (*Zhu et al., 2021*).

Another innovative approach proposes the use of spatial attention and channel mechanisms to improve the accuracy of small vessel detection without significantly increasing computational resources (*Munin et al., 2023*). This method also uses an innovative loss function to strengthen model training. The results show superior performance compared to other commonly used approaches.

For ship detection in synthetic aperture radar (SAR) images, the Duplicate Bilateral YOLO (DB-YOLO) is presented, providing real-time performance while maintaining high accuracy (*Loran et al., 2022*). This method uses a single-stage network and a duplicate bilateral feature pyramid to handle ship diversity in SAR images, achieving impressive results on two SAR ship datasets.

Finally, one study focuses on the identification and classification of ships to control marine pollution (*Ke et al., 2021*). The YOLOv8 model is evaluated for its ability to detect and classify vessels in real time. The results are very promising with a mAP@50 of 98.9%, but it is worth noting high GPU consumption. These findings are essential for the development of marine pollution control technologies and have a significant impact on the maritime industry, policy makers and researchers engaged in environmental protection.

Ship detection using Faster R-CNN has seen significant advancements in various maritime applications. Traditional systems such as AIS and marine radars are insufficient, hence the importance of airborne radars in the deep sea. To improve the accuracy, (*Li, Zhang & Wang, 2020*) introduced deformable convolution kernels in Faster R-CNN to model efficiently geometric transformation of ships. This improvement led to an increase in average accuracy of 2.02%.

*Gong et al. (2023)* proposed an approach for a speed of detection is crucial, especially in emergency situations. An innovative approach was proposed, including a light-weight baseline network, a scale selection method based on K-Means, and the use of RoIAlign. This method achieved an average accuracy of 0.898, 2.78% better than the classic Faster R-CNN, with 800% faster detection speed.

*Li, Qu & Shao (2017)* present a synthetic aperture radar images challenges due to the small size of the targets. The SSPNet network was developed with specific modules for the detection of small targets, resulting in an average accuracy (AP50) of 91.57%.

*Wei et al. (2020)* show that the use of deep learning for SAR vessel detection is growing. A dataset was created to evaluate the algorithms, and strategies such as feature fusion and transfer learning were adopted. The results show better accuracy and increased efficiency.

Remote sensing imagery is also an area of interest, where image pre-processing and the introduction of dilated convolution in Faster R-CNN have improved vessel detection

(*Maity, Banerjee & Sinha Chaudhuri, 2021*). An integrated software platform facilitates training of the vessel detection network.

Finally, river vessel detection was addressed with a method based on a regional convolutional neural network (*Chao et al., 2018*). This approach reduced false alarms and improved accuracy.

These advances demonstrate the growing importance of Faster R-CNN and deep learning for ship detection in a variety of maritime and remote sensing environments. Continuous improvements are needed to ensure safety at sea and improve management of marine resources. Faster R-CNN and YOLO (You Only Look Once) are two of CNN's most influential architectures for object detection. Faster R-CNN introduces the region proposal network (RPN) to generate potential regions of interest, improving detection efficiency and accuracy. YOLO, on the other hand, pro-poses a one-step approach that simultaneously predicts bounding boxes and object classes in a single network pass, enabling real-time detections.

This section of the literature highlights the importance of object detection at sea and sets the scene for our comparative study of Faster R-CNN and YOLOv8 in this specific context.

## METHODOLOGY

### Description of maritime monitoring data used

The maritime monitoring data used in this study is essential to evaluate the performance of the Faster R-CNN and YOLOv8 methods (*Terven & Cordova-Esparza, 2023*) in detecting fishing vessels and fish schools at sea. This data includes a collection of images and of video footage captured in various marine environments, covering a variety of scenarios and conditions. They were obtained from sources such as maritime surveillance cameras, drones and satellites. The diversity of these data reflects the complexity of the marine environment and ensures rigorous evaluation of model performance.

### Faster R-CNN overview

Faster R-CNN, based on CNN as shown on Fig. 1, is a two-stage object detection model. In its first step, it uses a RPN to generate potential regions of interest. In the second step, a convolutional neural network performs classification and refinement of the bounding boxes of the detected objects. This architecture allows precise localization and classification of objects in a single approach as proposed by *Zhang et al. (2022)*.

### YOLOv8 overview

Figure 2 shows the YOLOv8 architecture, an acronym for "You Only Look Once version 8," is an object detection model that is characterized by its one-step approach. It divides the image into a grid and simultaneously predicts the classes and bounding boxes of objects in each grid cell. YOLOv8 (*Ultralytics, 2024*) is known for its speed of execution and its ability to handle real-time detections.

*Affes et al. (2023)* proposed a comparison between different YOLO implementation for an intelligent video surveillance.

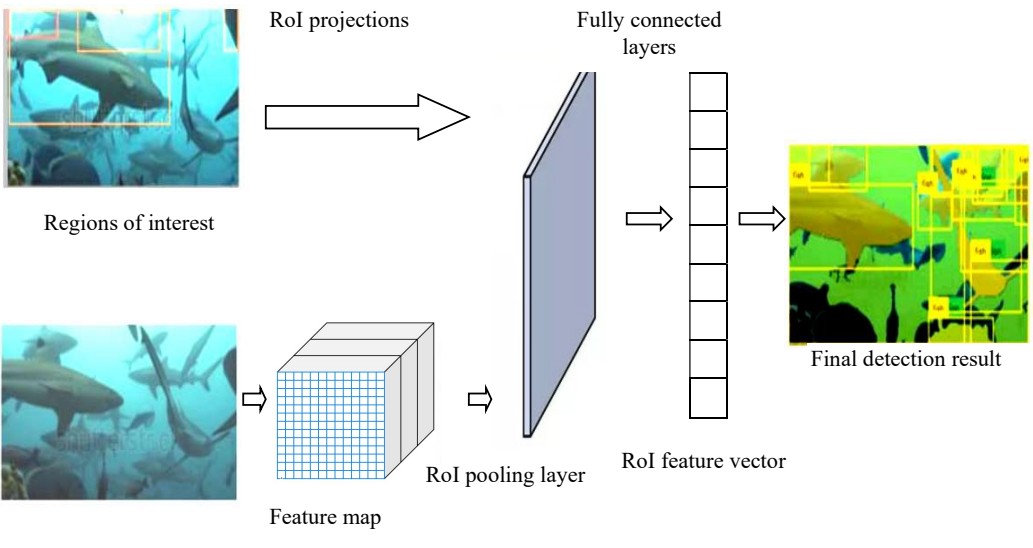

**Figure 1  Architecture of fish detection and recognition using Faster R-CNN.** Image taken from https://www.youtube.com/watch?v=W-Tccnlejcs.

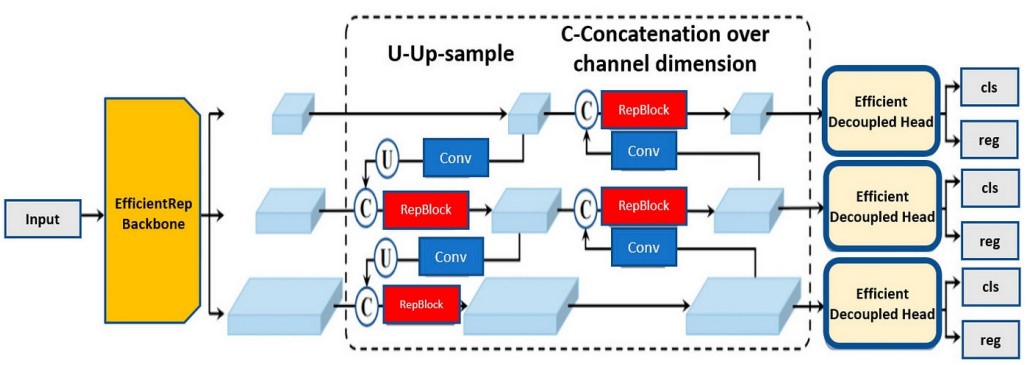

**Figure 2  Architecture of YOLOv8 (*Hussain, 2023*).**

## Data preprocessing and dataset preparation

The maritime surveillance data underwent an exhaustive pre-processing process to ensure data quality and adapt it to the requirements of Faster R-CNN and YOLOv8 models. The process includes image normalization to ensure consistent brightness and contrast levels, as well as noise reduction to improve the visual clarity of complex ocean scenes. Additionally, target objects, including fishing boats and fish, were annotated with extreme precision through the use of Autodistill, a computer-assisted annotation method that facilitates high-quality annotation. Additionally, the image database has been expanded to include a variety of marine scenes and conditions, ensuring a more complete representation of the potential challenges models face when detecting objects in marine environments.

**Table 1  Faster R-CNN model performance metrics.**

|  | Precision | Recall | F1 score | mAP |
|---|---|---|---|---|
| Fish's detection | 0.68 | 0.58 | 0.51 | 0.16 |
| Vessel's detection | 0.76 | 0.73 | 0.57 | 0.09 |

The use of the dataset for the different algorithms tested was as follows: 8.78% (63 images) of images for the test, 71.70% (448 images) for the train and 19.50% for the validation (128 images).

## Performance evaluation

The performance evaluation of the models was carried out using standard metrics such as precision, recall and F-measure. The execution time required for real-time detections in various maritime surveillance scenarios was recorded. The performance of computer vision models can be measured using a metric called average precision (mAP). It is equal to the average of the average precision measures for all classes of the model and provides a baseline measure for comparing different versions of the same model, taking into account both precision and recall (*Affes et al., 2023*). mAP is a metric that ranges from 0 to 1, and when a model has a high mAP, it means it has a lower false negative rate and a lower false positive rate. Due to the direct relationship, the precision and recall of the model will increase as the amplitude of the mAP increases. The detailed results of these evaluations are presented in the following section for in-depth analysis

## RESULT AND DISCUSSIONS

### Faster R-CNN model results

Table 1 below represents the fish and fishing vessel detection results with the various performance metrics, including precision, recall, F1 score and mean average precision (mAP).

Evaluation of the performance of fish and fishing vessel classification models re-veals significant difficulties. Accuracy for fish ranged from 0% to 45%, indicating limited ability to accurately predict different classes of fish. For fishing boats, the results were slightly better, with an accuracy of 62%, but still poor overall. Furthermore, the recall rates for both categories were equally unsatisfactory, ranging from 0% to 57% for fish and from 0% to 52% for fishing vessels, highlighting the gap in true positive detections. The F1 values for these two categories show an unstable balance between precision and recall, reaching 51% for fish and 57% for fishing boats, which is still not enough. Furthermore, the mAP for fish is estimated to be 0.16, indicating that the overall quality of predictions for different fish categories is relatively low, while the mAP for fishing vessels is even lower at 0.09, indicating that the overall quality of predictions for different fish categories is relatively poor.

Table 2 shows the fish detection confusion matrix, while Table 3 describes the fishing vessel detection confusion matrix. Confusion matrices provide crucial information about the performance of the fish and fishing vessel detection model. In the confusion matrix for

**Table 2  Fish detection confusion matrix.**

| True positive \\ Prediction | Creatures | fish | jellyfish | penguin | puffin | shark | starfish | stingray |
|---|---|---|---|---|---|---|---|---|
| creatures | 0 | 0 | 0 | 0 | 0 | 0 | 0 | 0 |
| fish | 0 | 143 | 62 | 8 | 3 | 19 | 3 | 11 |
| jellyfish | 0 | 108 | 23 | 6 | 7 | 5 | 2 | 3 |
| penguin | 0 | 30 | 15 | 32 | 1 | 4 | 0 | 0 |
| puffin | 0 | 5 | 9 | 1 | 8 | 2 | 0 | 0 |
| shark | 0 | 16 | 10 | 4 | 1 | 4 | 0 | 3 |
| starfish | 0 | 6 | 3 | 0 | 0 | 1 | 1 | 0 |
| stingray | 0 | 9 | 2 | 1 | 0 | 2 | 0 | 1 |

fish detection, the largest values include the high number of false negatives in categories fish, jellyfish, and shark, indicating that the model has difficulty identifying these specific classes of fish. Similarly, for the detection of fishing vessels, the significant values lie in the high false positives in categories 3, 4, and 6, indicating frequent confusions between these classes of fishing vessels. The high numbers in the diagonal cells of both matrices indicate relatively adequate ability to predict some classes, although improvements are needed to reduce classification errors and increase overall precision and recall.

## YOLOv8 model results

Before presenting Table 4 below, it is important to note the remarkable performance of the YOLOv8 model in detecting fish and fishing vessels. For fish, the model achieved a precision of 75.97%, a recall of 60.72%, and an mAP50 of 70.91%. In the case of fishing vessels, the model demonstrated exceptional precision of 87.42%, recall of 90.04%, and mAP50 and mAP50-95 of 93.97% and 91.20% respectively. These performances demonstrate the effectiveness of the YOLOv8 model in both detection domains.

These results indicate the reliability of the YOLOv8 model for detecting both fish and fishing vessels, highlighting its ability to provide accurate predictions in both of these object detection scenarios. Figure 3 illustrates performance metrics for fish detection while Fig. 4 describes performance metrics for detection of fish vessels.

Performance analysis of the YOLOv8 model for detecting fish and fishing vessels highlights impressive results. In the case of fish, the model displays an accuracy of 75.97%,

Table 3  Fishing vessel detection confusion matrix.

| True positive \ Prediction | Ships | 0 | 1 | 2 | 3 | 4 | 5 | 6 | 7 | 8 | 9 |
|---|---|---|---|---|---|---|---|---|---|---|---|
| Ships | 0 | 0 | 0 | 0 | 0 | 0 | 0 | 0 | 0 | 0 | 0 |
| 0 | 0 | 16 | 0 | 85 | 15 | 3 | 24 | 6 | 8 | 3 | 0 |
| 1 | 0 | 1 | 0 | 16 | 2 | 1 | 3 | 0 | 0 | 0 | 0 |
| 2 | 0 | 51 | 3 | 347 | 84 | 27 | 81 | 18 | 26 | 22 | 4 |
| 3 | 0 | 9 | 4 | 51 | 13 | 4 | 16 | 4 | 1 | 2 | 3 |
| 4 | 0 | 2 | 0 | 10 | 2 | 1 | 4 | 0 | 1 | 1 | 0 |
| 5 | 0 | 3 | 0 | 8 | 1 | 0 | 0 | 1 | 0 | 1 | 0 |
| 6 | 0 | 4 | 0 | 15 | 6 | 0 | 5 | 1 | 1 | 1 | 0 |
| 7 | 0 | 0 | 0 | 0 | 0 | 0 | 0 | 0 | 0 | 0 | 0 |
| 8 | 0 | 3 | 1 | 28 | 2 | 3 | 8 | 0 | 1 | 2 | 0 |
| 9 | 0 | 0 | 0 | 4 | 0 | 0 | 0 | 1 | 0 | 1 | 0 |

Table 4  YOLOv8 model performance metrics.

| | Precision | Recall | mAP50 | mAP50-95 |
|---|---|---|---|---|
| Fish's detection | 0.7597 | 0.6072 | 0.7091 | 0.3741 |
| Vessel's detection | 0.87423 | 0.90039 | 0.93971 | 0.91199 |

meaning that nearly 76% of the fish detected are correctly identified. The recall of 60.72% indicates that the model manages to capture more than half of the fish present in the images. The mAP50 of 70.91% underlines the effectiveness of the model in maintaining a high average precision over 50% of the confidence thresholds, despite the complexity of the task. Regarding fishing vessels, the model demonstrates an exceptional accuracy of 87.42%, highlighting its ability to minimize false detections. A recall of 90.04% indicates that the model identifies the vast majority of real fishing vessels. The values of mAP50 (93.97%) and mAP50-95 (91.20%) highlight the model's ability to achieve exceptional performance across various confidence thresholds. These results demonstrate the effectiveness and reliability of the YOLOv8 model in accurately detecting fish and fishing vessels, making it a powerful tool for maritime surveillance applications.

## DISCUSSION

In our context, the superiority of the YOLOv8 model over Faster R-CNN in this study can be attributed to several factors that should have been explicitly discussed in the article.

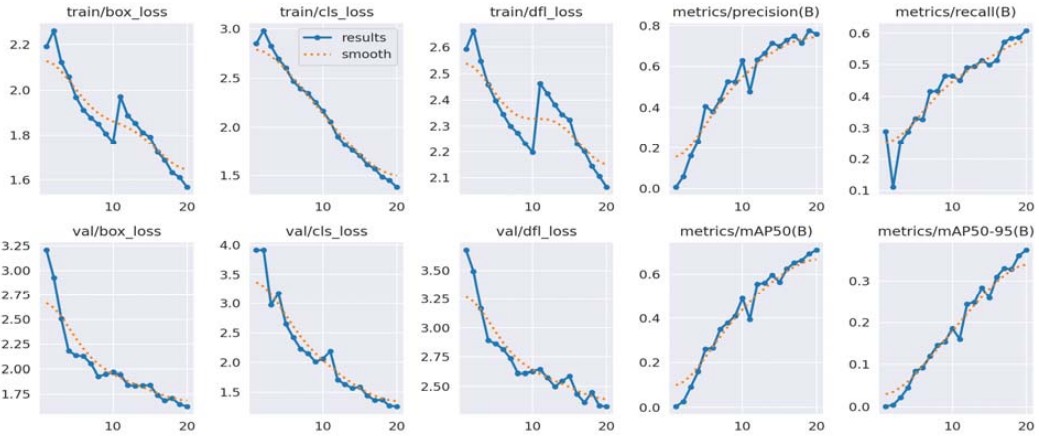

**Figure 3** Performance metrics for fish detection.

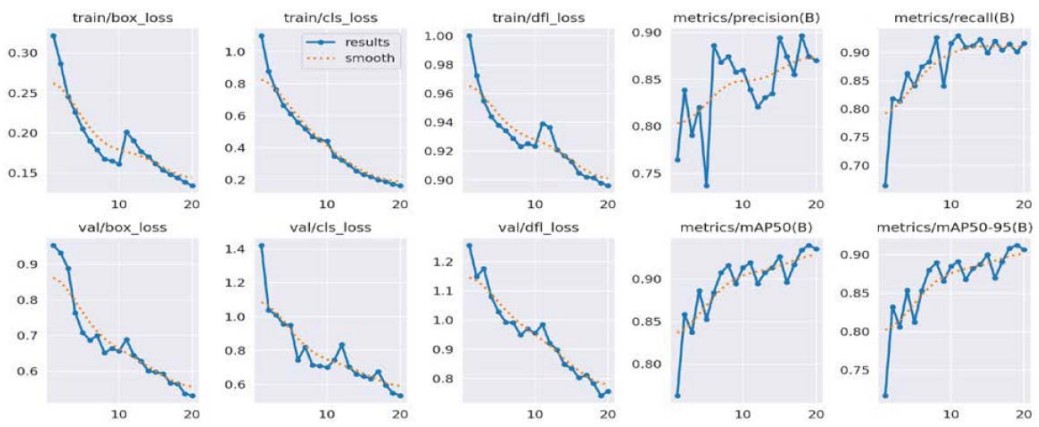

**Figure 4** Performance metrics for detection of fishing vessels.

Firstly, YOLOv8's one-step approach, which simultaneously predicts bounding boxes and object classes, enables real-time detections and may result in faster and more accurate object localization compared to Faster R-CNN's two-stage approach. Additionally, YOLOv8 is known for its speed of execution and its ability to handle real-time detections, which could contribute to its superior performance in detecting objects in dynamic maritime environments. Furthermore, the architectural differences between the two models, such as the presence of the region proposal network (RPN) in Faster R-CNN, may affect their respective performances in detecting objects with varying scales and aspect ratios. These factors, along with the specifics of the dataset and training process, could explain why YOLOv8 outperformed Faster R-CNN in this study.

The results obtained with Faster R-CNN and YOLOv8 models for fish and fishing vessel detection highlight the comparative performance. Although the Faster R-CNN model shows some precision, it seems difficult to achieve the best balance between precision

and recall: the scores for fish are 68% and 58%, respectively, and the scores for fishing vessels are respectively 76% and 71%. In comparison, the YOLOv8 model performed well, with accuracy of 76% for fish and 87% for fishing vessels, with recall rates of 61% and 90% respectively. This superior performance of the YOLOv8 model is also confirmed by the mAP50 values of 71% for fish and 94% for fishing boats, outperforming the Faster R-CNN model. To improve these models, it is necessary to invest in deeper training and more diverse and higher quality datasets. Exploring data augmentation techniques as well as hyperparameter optimization can also help improve the performance of these models. Furthermore, adopting the latest models with improved architecture could be an avenue to explore to improve the efficiency of fish and fishing vessel detection in different environments.

## CONCLUSIONS

Our comparative study of Faster R-CNN and YOLOv8 models for fish and fishing vessel detection highlights the crucial importance of artificial intelligence in maritime surveillance. The results obtained significantly demonstrate the superiority of the YOLOv8 model in both detection domains. The Faster R-CNN model struggles to find a balance between precision and recall, with scores of 68% and 58% respectively for fish and 76% and 71% respectively for fishing boats, while YOLOv8 shows a excellent accuracy, with scores of 76% for fish and 76% for vessels, and 87% are fishing boats. The mAP50 values of 71% for fish and 94% for fishing boats in YOLOv8 are also better than those in Faster R-CNN.

These findings highlight the effectiveness and reliability of the YOLOv8 model in accurately detecting fish and fishing vessels, making it a powerful tool for maritime surveillance applications. To improve these models, it is crucial to invest in deeper training using diverse, high-quality datasets. Techniques such as data augmentation and hyperparameter optimization can also fine-tune its performance. Additionally, exploring state-of-the-art models with improved architectures can improve the efficiency of detecting fish and fishing vessels in various marine environments. These results highlight the potential of advanced deep learning models such as YOLOv8 to revolutionize maritime surveillance applications, paving the way for future technologies that will play a central role in protecting marine resources and marine ecosystems.

It is important to note that in our work we not only dealt with fish, but also with boats. To this end, we have used not only fish images but also other boat images supplied by our partner. As part of the perspectives of this work, we will test our algorithm on proposed databases such as DeepFish and OzFish.

### Funding
This research has been funded by Scientific Research Deanship at University of Ha'il, Saudi Arabia through project number RG-23 088. The funders had no role in study design, data collection and analysis, decision to publish, or preparation of the manuscript.

## Grant Disclosures

The following grant information was disclosed by the authors:
Scientific Research Deanship at University of Ha'il, Saudi Arabia: RG-23 088.

## Competing Interests

The authors declare there are no competing interests.

## Author Contributions

- Lotfi Ezzeddini conceived and designed the experiments, performed the computation work, prepared figures and/or tables, and approved the final draft.
- Jalel Ktari conceived and designed the experiments, performed the computation work, authored or reviewed drafts of the article, and approved the final draft.
- Tarek Frikha conceived and designed the experiments, analyzed the data, authored or reviewed drafts of the article, and approved the final draft.
- Naif Alsharabi performed the experiments, prepared figures and/or tables, and approved the final draft.
- Abdulaziz Alayba analyzed the data, authored or reviewed drafts of the article, and approved the final draft.
- Abdullah J. Alzahrani analyzed the data, authored or reviewed drafts of the article, and approved the final draft.
- Amr Jadi performed the computation work, prepared figures and/or tables, and approved the final draft.
- Abdulsalam Alkholidi conceived and designed the experiments, prepared figures and/or tables, and approved the final draft.
- Habib Hamam performed the experiments, authored or reviewed drafts of the article, and approved the final draft.

## Data Availability

The Ship and Fast RCNN Implementation are available at GitHub and Zenodo:

- https://github.com/lotfiezdini/ship-detection-faster-cnn

- ktari, . jalel ., frikha, . tarek ., & Alsharabi, N. (2024). ship-detection-faster-cnn-main. Zenodo. https://doi.org/10.5281/zenodo.10992875

- https://github.com/lotfiezdini/Ship-detection-YOLOv8

- ktari, . jalel ., frikha, . tarek ., & A. Alsharabi, N. (2024). Ship-detection-YOLOv8-main [Data set]. Zenodo. https://doi.org/10.5281/zenodo.10992851

The fish YOLO code is available in the Supplemental Files.

## Supplemental Information

Supplemental information for this article can be found online at http://dx.doi.org/10.7717/peerj-cs.2033#supplemental-information.

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
