# Peer review of "Analysis of the performance of Faster R-CNN and YOLOv8 in detecting fishing vessels and fishes in real time"

_PeerJ Computer Science, doi:10.7717/peerj-cs.2033_

## Round 0.1 · original submission · Major Revisions

Dear authors,

You are advised to critically respond to all comments point by point when preparing a new version of the manuscript and while preparing for the rebuttal letter. Please address all the comments/suggestions provided by the reviewers.

Reviewer 1 has suggested that you cite specific references. You are welcome to add it/them if you believe they are relevant. However, you are not required to include these citations, and if you do not include them, this will not influence my decision.

Reviewer 2 has pointed to the need for more manuscript structural and investigation design

Kind regards,
PCoelho

**Language Note:** The review process has identified that the English language must be improved. PeerJ can provide language editing services - please contact us at [email protected] for pricing (be sure to provide your manuscript number and title). Alternatively, you should make your own arrangements to improve the language quality and provide details in your response letter. – PeerJ Staff

Reviewer 1 ·

Basic reporting

The abstract should summarize the research, highlighting the focus on comparing the performance of Faster R-CNN and YOLOv8 in real-time detection of fishing vessels and fishes. Emphasize the significance of the study, its potential impact on fisheries monitoring, and the contribution to the field of object detection using deep learning.
Clearly state the research objectives, specifically comparing the performance of Faster R-CNN and YOLOv8. Highlight the relevance of real-time detection in fisheries management and the challenges addressed by the study.
Conduct a comprehensive literature review to establish the current state-of-the-art in object detection, particularly in the context of fisheries monitoring. Discuss existing methods, challenges, and limitations in real-time detection of fishing vessels and fishes.

The following article can be included in the discussion
Statistical analysis of design aspects of various YOLO-based deep learning models for object detection.
Identify gaps in the literature that the current research aims to fill and emphasize the uniqueness of comparing Faster R-CNN and YOLOv8.

Experimental design

Explain the selection criteria for the models, the dataset used (including its source and characteristics), and the evaluation metrics employed.
Address the strengths and weaknesses of Faster R-CNN and YOLOv8 in real-time detection, considering factors such as accuracy, processing speed, and scalability.

Validity of the findings

Present the results of the comparative analysis between Faster R-CNN and YOLOv8 in detecting fishing vessels and fishes in real-time. Provide quantitative data on key performance metrics, including accuracy, precision, recall, and processing speed.

Additional comments

Summarize the key findings and contributions of the research. Emphasize the significance of comparing Faster R-CNN and YOLOv8 in real-time fisheries monitoring. Discuss the practical implications of the results and offer recommendations for future research, such as exploring additional models, improving dataset diversity, or integrating other sensors for enhanced detection.

·

Basic reporting

The manuscript could be improved with language editing (grammar, technical writing and proofing). The introduction is very brief. It should contain a general background followed by a critic of the literature and the contributions made in this manuscript. The datasets, which are of specific interest is not clearly mentioned if collected by them to taken from others, a loose reference is given [24] which is a review paper. The datasets are not shared either.

Experimental design

Only mAP is used. There are other metrics that are widely used in the literature are not being employed here. Also it is not clear if the two models were trained for the data, or it is just application of the models. Why only two models are selected?

Validity of the findings

The discussion is missing. Why one method is working better should have been explained.

---

## Round 0.2 · Major Revisions

Dear authors,

You are advised to critically respond to all comments point by point when preparing a new version of the manuscript and while preparing for the rebuttal letter.
Reviewer #2 states that does not address the concerns shown in the first round of review. Please address all the comments/suggestions provided by the reviewers.

Kind regards,
PCoelho

Reviewer 1 ·

Basic reporting

No Comment

Experimental design

No Comment

Validity of the findings

No Comment

Additional comments

No Comment

·

Basic reporting

The revision does not address the concerns showed in the first round of review. The dataset is shared to the reviewer in a private google drive repository. It is recommended that the dataset and the sources are shared via some permanent repositories. Also, the manuscript does not contain any link or revised description on the dataset. The comments on the restructuring of the relevant section is not done. There are number of publicly available datasets like DeepFish, OzFish, etc and on which sota methods have been tried. The paper should make comparisons on those datasets and with those methods. Many of the figures, including the confusion matrices are unreadable. Contributions are limited and not clearly mentioned.

Experimental design

Many of the important works are missing. The dataset description is missing.

Validity of the findings

The figures are not clear. Also comparisons are not made to sota methods.

---

## Round 0.3 · accepted · Accept

Dear authors, we are pleased to verify that you meet the reviewer's valuable feedback to improve your research.

Thank you for considering PeerJ Computer Science and submitting your work.

Reviewer 1 ·

Basic reporting

good

Experimental design

good

Validity of the findings

good

·

Basic reporting

The authors seem to not responding to the previous comments. This time reference to dataset was given in literature review section. No comparisons are made. Organizational structure remains the same as in the the first submission.

Experimental design

No comparisons are made.

Validity of the findings

Major concerns

Reviewer 3 ·

Basic reporting

The paper presents a comparative study aimed at evaluating the efficacy of two deep learning models, Faster R-CNN and YOLOv8, in maritime surveillance. The significance of the research, well-backed by a robust literature review, sets a solid groundwork for understanding the advancements in real-time object detection technologies in challenging marine environments.

Experimental design

The paper describes a comparative analysis between two deep learning models—Faster R-CNN and YOLOv8—for detecting fishing vessels and fish in maritime surveillance scenarios. The study evaluates these models based on metrics like accuracy, execution speed, and robustness to environmental conditions.

Validity of the findings

The study's findings are supported by a methodical approach and rigorous metrics